# Demystifying *Cassiopea* species identity in the Florida Keys: *Cassiopea xamachana* and *Cassiopea andromeda* coexist in shallow waters

**Kaden Muffett** ⓘ *, **Maria Pia Miglietta** ⓘ

Texas A&M University at Galveston, Galveston, Texas, United States of America

* kmmuffett@gmail.com

## Abstract

The phylogeny of the Upside-Down Jellyfish (*Cassiopea* spp.) has been revised multiple times in its history. This is especially true in the Florida Keys, where much of the *Cassiopea* stock for research and aquarium trade in the United States are collected. In August 2021, we collected 55 *Cassiopea* medusae at eight shallow water sites throughout the Florida Keys and sequenced *COI*, *16S*, and *28S* genes. Mitochondrial genes demonstrate that the shallow waters in Florida are inhabited by both *Cassiopea xamachana* and a non-native *Cassiopea andromeda* lineage, identified in multispecies assemblages at least thrice. While *C. xamachana* were present at all sites, the *C. andromeda*-mitotype individuals were present at only a minority of sites. While we cannot confirm hybridization or lack thereof between the *C. xamanchana* and *C. andromeda* lineages, these previously unknown multispecies assemblages are a likely root cause for the confusing and disputed *COI*-based species identities of *Cassiopea* in the Florida Keys. This also serves as a cautionary note to all *Cassiopea* researchers to barcode their individuals regardless of the location in which they were collected.

**Editor:** Sergio N. Stampar, Sao Paulo State University Julio de Mesquita Filho Bauru Campus Faculty of Sciences: Universidade Estadual Paulista Julio de Mesquita Filho Faculdade de Ciencias Campus de Bauru, BRAZIL

## Introduction

Marine invertebrates have a wealth of cryptic lineages [1, 2]. Although our understanding of the diversity within cryptic marine taxa has grown, precise distributions of species are often challenging to assess with common sampling approaches. Scyphozoan phylogenies have frequently been sampled across large geographic areas but with few individuals at each site [3, 4]. In some pelagic systems, such as *Aurelia* or *Chrysaora*, this approach may be sufficient [5, 6]. However, in systems with a strong invasion potential and limited natural dispersion capabilities, like the genus *Cassiopea*, this shallow or single-site sampling may result in inadequate coverage to identify all species present.

As in other scyphozoan lineages, *Cassiopea* suffers from poor phylogenetic clarity. This is consequential because *C. xamachana*, and the genus *Cassiopea* more broadly, have become an

**Data Availability Statement:** All relevant data and accession numbers are within the manuscript and its Supporting Information files.

**Funding:** This work was supported by a Texas Sea Grant to author KMM. This work was also supported by Texas A&M University Galveston. The funders had no role in study design, data collection and analysis, decision to publish, or preparation of the manuscript.

**Competing interests:** The authors have declared that no competing interests exist.

emergent model system for research in symbiosis, behavior, and regeneration [7–11]. Recent work has demonstrated that morphological, symbiosis and ecological differences exist between *Cassiopea* species [12, 13], however, most non-taxonomic research is done on unspecified *Cassiopea* [14, 15]. This lack of clarity on the identity of the *Cassiopea* used in research, is problematic as it may lead to confounded, unreproducible, or less comparable experimental results.

High morphological heterogeneity within populations and apparent crypsis across species make *Cassiopea* difficult to identify [3, 8, 12]. *Cassiopea* species identification and species boundaries are even more blurred in the Florida Keys, one of the main collection grounds for *Cassiopea* research in the United States [12]. *Cassiopea* from Florida are often arbitrarily assigned one of three names: *C. xamachana*, *C. andromeda* or *C. frondosa*. *C. xamachana*, originally described in Jamaica by Bigelow 1892, theoretically represents the dominant morphotype in the Florida Keys and Caribbean [16]. *C. xamachana*'s description is distinct from *C. andromeda* (Forskål, 1775), a Red Sea native and an invasive documented in Brazil, the Mediterranean, and Hawai'i, though many of the characters on which that distinction was made are variable [3, 17, 18]. The separation of *C. xamachana* and *C. andromeda* is a point of contention, and the two species have been sometimes considered synonymous. When synonymized, *C. xamachana* was considered a population of introduced *C. andromeda* [8, 19]. This synonymization was originally a product of Gohar and Eisawy's efforts to reduce the many described *Cassiopea* species into only three species—*C. dieuphila* (species name not revisited), *C. andromeda* and *C. frondosa*—based solely on one morphological character, the rhopaliar number [19]. Rhopaliar number is now recognized as having high intraspecific variability and limited interspecific variability and thus not suitable for species delimitation [3, 13]. Four decades later, genetic sampling of *Cassiopea* by Holland et al. (2004) across the globe upended this oversimplistic morphological assessment and significantly expanded the known species diversity in the genus but found no evidence of a "*C. xamachana*" lineage. The "*C. xamachana*" individuals (n = 4) collected from Bermuda and the Florida Keys by Holland et al. 2004 were not distinct from other global *C. andromeda* populations [3]. The *C. frondosa* species, a distinctive deeper water *Cassiopea*, and a more distant relative also found within the Keys, has remained a stable clade throughout various phylogenies and as such we did not sample these medusae [12] Since Holland et al. (2004) [3], some of the *Cassiopea* species invalidated by Gohar and Eisawy (1960) were validated and characterized morphologically, however the *C. xamachana*/*C. andromeda* distinction was not reevaluated [13], as such we have not met the need for an updated and more in-depth phylogeny of the shallow water Florida Keys *Cassiopea*.

Here, using a concerted sampling of eight shallow water sites in the Florida Keys, and using mitochondrial and nuclear genes, as well as morphological data, we address the presence and the phylogenetic status of *C. xamachana* and *C. andromeda* in the Florida Keys.

## Materials and methods

### Collection

In August 2021, 55 *Cassiopea* were collected by hand in 8 sites along the length of the Florida Keys. All sites were from near-shore water under 2 m depth (S1 Table in S1 File). Each *Cassiopea* was photographed, diameter measured, and small tissue samples were preserved in ethanol (95%) at room temperature. Six small individuals were fully preserved in 95% EtOH, these samples were later used for limited morphological analysis (S1 Table in S1 File). Medusa density was estimated in area of collection by marking out one square meter with a tape measure and counting medusae within the area. Permitting for these specimens was waived by Florida Fish and Wildlife Conservation Commision.

## DNA sequencing

DNA extractions from the 55 specimens were performed according to a salting out protocol (see full protocol in [20]). Mitochondrial Cytochrome c oxidate subunit I (*COI*) and *16S* ribosomal RNA (*16S*) were amplified using *Cassiopea*-specific protocols and primers [13]. Nuclear *28S* was amplified using the *Cassiopea*-specific primers and protocols in Daglio et al. 2017. All products were purified using ExoSap and sanger sequenced at Texas A&M-Corpus Christi Genomics Core (Corpus Christi, Texas) or GeneWiz Azenta (Plainsfield, New Jersey). Fifty-five 514 bp segments of *COI*, 34 575 bp segments of *16S* and 18 822 bp segments of *28S* were assembled on Geneious and checked by eye.

## Data analysis

***C. xamachana* and *C. andromeda* clade validation dataset.**   Newly produced *COI* and *16S* sequences (55 and 38 sequences respectively) were aligned using Geneious 2022.2.2 (https://www.geneious.com) to *COI* and *16S* sequences from the *C. xamachana* and *C. andromeda* published genomes. Provisional sequence identity was assigned through agreement >98% to either the *C. xamachana* genome [21] or the *C. andromeda* mitogenome [22]. All sequences were uploaded to Genbank (see S1 Table in S1 File for complete list of accession numbers).

**Combined *COI* and *16S Cassiopea* phylogeny dataset.**   A subset of 11 representative *COI* sequences from individuals collected in this study (see Table 1 and S1 Table in S1 File) were aligned with *Cassiopea COI* GenBank sequences from Holland et al. 2004, Morandini et al. 2016 [17], Daglio et al. 2017, Abboud et al. 2018, Gamero-Mora et al. 2022, Kayal et al. 2013, and *Mastigias* and *Versuriga* from Swift et al. 2016 and Sun et al. 2019 (see table) using MAFFT 7 (L-INS-i) [22–25]. The final *COI* dataset was composed of 89 sequences, 11 of which were produced here, all trimmed to 514bp. The remaining 78 GenBank sequences belonged to the following species: *C. andromeda*, *C. xamachana*, *C. ornata*, *C.* sp. 1, *C.* sp. 2, *C.* sp. 3, *C. culionensis*, *C. mayeri*, *C. frondosa*, and outgroups *Mastigias papua* and *Versuriga anadyomene*.

*16S* sequences from 10 representative individuals across the Florida Keys (see Table 1) collected in this study were aligned with *Cassiopea 16S* GenBank sequences from Gamero-Mora 2022 and Daglio et al. 2017 and trimmed to 544 bp. The final *16S* dataset was composed of 22 sequences, ten from *C. xamachana* and *C. andromeda* from this study (same individuals as *COI* minus one) and 12 GenBank sequences belonging to *C. andromeda*, *C. xamachana*, *C. culionensis*, *C. mayeri*, *C. frondosa*, *C. ornata*, and outgroup taxa *M. papua* and *V. anadyomene*. *16S* sequences were aligned using MAFFT (E-INS-i).

Models for all *COI* codons and *16S* dataset were chosen by AICc from MEGA X 11 [26] model tester (*COI* codon position 1: TN93+I, *COI* codon position 2: TN93+I, *COI* codon position 3: TN93+G+I, *16S*: GTR+G+I). A dataset combining *COI* and *16S* was run in IQtree 1.6 under an ML framework with support from 1000 aLRT (approximate likelihood ratio test) and 1000 non-parametric bootstraps. Bayesian support was generated in BEAST 1.8 [27] with 10 million steps. All phylogenetic trees were edited with Figtree v1.4.4 (http://tree.bio.ed.ac.uk/software/figtree/).

***28S* dataset.**   The *28S* dataset consisted of 28 sequences—18 newly produced *Cassiopea* sequences, 8 Genbank sequences of *C. frondosa*, *C. andromeda*, *C. ornata*, and two sequences belong to outgroups *M. papua* and *V. anadyomene* (see S2 Table in S1 File for all accession numbers). All 822 to 846 nucleotide *28S* sequences were aligned using MAFFT (G-INS-i). *28S* phylogenetic trees were run on IQtree under ML framework with support from aBayes, 1000

**Table 1.** *COI* and *16S* sequences used for combined tree generation, with both original identity from source and identity post-analysis.

| Species | Reported species upon sequence publication | Locality | COI Accession | 16S Accession | Dataset Used in | Source |
|---|---|---|---|---|---|---|
| *C. andromeda* | - | Cudjoe Key, FL, USA | OP503345 | OP503932 | *^+# | This study |
| *C. andromeda* | - | Key Largo, FL, USA | OP503353 | OP503938 | *^+# | This study |
| *C. andromeda* | - | Key West, FL, USA | OP503325 | OP503913 | *^+# | This study |
| *C. andromeda* | C. andromeda | Hilton lagoon, Waikiki, leeward O'ahu, Hawai'i, USA | AF231109 | - | *+ | Holland et al. 2004 |
| *C. andromeda* | C. andromeda | Hilton lagoon, Waikiki, leeward O'ahu, Hawai'i, USA | AY319448 | - | *+ | Holland et al. 2004 |
| *C. andromeda* | C. andromeda | Hilton lagoon, Waikiki, leeward O'ahu, Hawai'i, USA | AY319449 | - | *+ | Holland et al. 2004 |
| *C. andromeda* | C. andromeda | Hilton lagoon, Waikiki, leeward O'ahu, Hawai'i, USA | AY319450 | - | *+ | Holland et al. 2004 |
| *C. andromeda* | C. andromeda | Kainaone fish pond, Moloka'i, Hawai'i, USA | AY319453 | - | *+ | Holland et al. 2004 |
| *C. andromeda* | C. andromeda | Kainaone fish pond, Moloka'i, Hawai'i, USA | AY319454 | - | *+ | Holland et al. 2004 |
| *C. andromeda* | C. andromeda | El Ghardaqa, Egypt | AY319458 | - | *+ | Holland et al. 2004 |
| *C. andromeda* | C. andromeda | French Polynesia | JN700934 | JN700934 | *^+# | Kayal et al. 2013 |
| *C. andromeda* | C. andromeda | Brazil | KC464458 | - | *+ | Morandini et al. 2017 |
| *C. andromeda* | C. andromeda | Isla San José, Baja California Sur, Mexico | KY610551 | KY610609 | *^+# | Daglio et al. 2017 |
| *C. andromeda* | C. andromeda | Isla San José, Baja California Sur, Mexico | KY610552 | - | *+ | Daglio et al. 2017 |
| *C. andromeda* | C. andromeda | Isla San José, Baja California Sur, Mexico | KY610553 | - | *+ | Daglio et al. 2017 |
| *C. andromeda* | C. andromeda | Isla San José, Baja California Sur, Mexico | KY610554 | - | *+ | Daglio et al. 2017 |
| *C. andromeda* | C. andromeda | Isla San José, Baja California Sur, Mexico | KY610555 | - | *+ | Daglio et al. 2017 |
| *C. andromeda* | C. andromeda | Isla San José, Baja California Sur, Mexico | KY610556 | - | *+ | Daglio et al. 2017 |
| *C. andromeda* | - | Key Largo, FL, USA | OP503367 | OP503939 | *^+# | This study |
| *C. andromeda* | C. sp. | Walsingham Pond, Hamilton, Bermuda | MF742175 | - | *+ | Abboud et al. 2018 |
| *C. andromeda* | C. sp. | Mo'orea, Windward Islands, French Polynesia | MF742213 | - | *+ | Abboud et al. 2018 |
| *C. andromeda* | C. sp. | Mo'orea, Windward Islands, French Polynesia | MF742214 | - | *+ | Abboud et al. 2018 |
| *C. andromeda* | C. sp. | Mo'orea, Windward Islands, French Polynesia | MF742215 | - | *+ | Abboud et al. 2018 |
| *C. andromeda* | C. andromeda* | Walsingham Pond, Bermuda | AY319463 | - | *+ | Holland et al. 2004 |
| *C. andromeda* | C. andromeda* | Richardson Bay, Bermuda | AY319464 | - | *+ | Holland et al. 2004 |
| *C. andromeda* | C. andromeda* | Richardson Bay, Bermuda | AY319465 | - | *+ | Holland et al. 2004 |
| *C. andromeda* | C. andromeda* | Walsingham Pond, Bermuda | AY319466 | - | *+ | Holland et al. 2004 |
| *C. andromeda* | C. andromeda* | Key Largo, Florida, USA | AY319468 | - | *+ | Holland et al. 2004 |
| *C. xamachana* | C. frondosa | Bahia Delfines, Bocas del Toro, Panama | KY610557 | - | *+ | Daglio et al. 2017 |
| *C. xamachana* | C. frondosa | Bahia Delfines, Bocas del Toro, Panama | KY610558 | - | *+ | Daglio et al. 2017 |
| *C. xamachana* | C. frondosa | Bahia Delfines, Bocas del Toro, Panama | KY610559 | KY610614 | *^+# | Daglio et al. 2017 |
| *C. xamachana* | C. sp. | Key Largo, Florida, USA | MF742149 | - | *+ | Abboud et al. 2018 |

*(Continued)*

**Table 1.** (Continued)

| Species | Reported species upon sequence publication | Locality | COI Accession | 16S Accession | Dataset Used in | Source |
|---|---|---|---|---|---|---|
| *C. xamachana* | *C. sp.* | Cassiopea Lake, Koror State, Palau | MF742166 | - | *+ | Abboud et al. 2018 |
| *C. xamachana* | - | Marathon Key, FL, USA | OP503314 | OP503902 | *^+# | This study |
| *C. xamachana* | - | Tavernier, FL, USA | OP503334 | OP503922 | *^+# | This study |
| *C. xamachana* | - | Lobster Walk, Monroe County, FL, USA | OP503341 | OP503929 | *^+# | This study |
| *C. xamachana* | - | Cudjoe Key, FL, USA | OP503343 | OP503931 | *^+# | This study |
| *C. xamachana* | - | Big Pine Key, FL, USA | OP503317 | OP503907 | *^+# | This study |
| *C. xamachana* | - | Big Pine Key, FL, USA | OP503320 | - | *+ | This study |
| *C. xamachana* | - | Key West, FL, USA | OP503326 | OP503914 | *^+# | This study |
| *C. xamachana* | *C. xamachana* | Panama | JN700936 | JN700936 | *^+# | Kayal et al. 2013 |
| *C. xamachana* | *C. xamachana* | Bahia Delfines, Bocas del Toro, Panama | KY610560 | - | * | Daglio et al. 2017 |
| *C. xamachana* | *C. xamachana* | Bahia Delfines, Bocas del Toro, Panama | KY610561 | - | * | Daglio et al. 2017 |
| *C. xamachana* | *C. xamachana* | Bahia Delfines, Bocas del Toro, Panama | KY610562 | - | * | Daglio et al. 2017 |
| *C. culionensis* | *C. culionensis* | Monterey Bay Aquarium, USA | KF683387 | - | * | Mellas et al. 2014 |
| *C. culionensis* | *C. culionensis* | Philippines | MW160923 | MW164879 | *^ | Gamero-Mora 2022 |
| *C. culionensis* | *C. culionensis* | Philippines | MW160930 | MW164886 | *^ | Gamero-Mora 2022 |
| *C. frondosa* | *C. frondosa* | Key Largo, Florida, USA | AY319467 | KY610617 | *^ | Holland et al. 2004 and Daglio et al. 2017 |
| *C. frondosa* | *C. frondosa* | San Blas Islands, Panama | AY319469 | - | * | Holland et al. 2004 |
| *C. frondosa* | *C. frondosa* | San Blas Islands, Panama | AY319470 | - | * | Holland et al. 2004 |
| *C. mayeri* | *C. mayeri* | Japan | MW160931 | MW164859 | *^ | Gamero-Mora 2022 |
| *C. mayeri* | *C. mayeri* | Philippines | MW160934 | MW164863 | *^ | Gamero-Mora 2022 |
| *C. mayeri* | *C. sp.* | Sorido Bay, Kri, Papua | MF742205 | - | * | Abboud et al. 2018 |
| *C. ornata* | *C. ornate* | Short Drop Off, Palau | AY319456 | - | * | Holland et al. 2004 |
| *C. ornata* | *C. ornata* | Kakaban, Kalimantan, Indonesia | AY319472 | AB720918 | *^ | Holland et al. 2004 and Gamero-Mora 2022 |
| *C. ornata* | *C. ornate* | Kakaban, Kalimantan, Indonesia | AY319473 | - | * | Holland et al. 2004 |
| *C. ornata* | *C. sp.* | Risong Cove, Auluptagel Island, Koror State, Palau | MF742179 | - | * | Abboud et al. 2018 |
| *C. ornata* | *C. sp.* | Risong Cove, Auluptagel Island, Koror State, Palau | MF742193 | - | * | Abboud et al. 2018 |
| *C. sp. 3* | *C. sp. 3* | Kahuku windward, Oahu, Hawai'i, USA | AY319452 | - | * | Holland et al. 2004 |
| *C. sp.* | *C. sp.* | Coombabah Creek, Queensland, Australia | MF742133 | - | * | Abboud et al. 2018 |
| *C. sp.* | *C. sp.* | Coombabah Creek, Queensland, Australia | MF742135 | - | * | Abboud et al. 2018 |
| *C. sp.* | *C. sp.* | Kakaban, Berau, Kalimantan Timur, Indonesia | MF742139 | - | * | Abboud et al. 2018 |
| *C. sp.* | *C. sp.* | Kakaban, Berau, Kalimantan Timur, Indonesia | MF742140 | - | * | Abboud et al. 2018 |
| *C. sp.* | *C. sp.* | Kakaban, Berau, Kalimantan Timur, Indonesia | MF742141 | - | * | Abboud et al. 2018 |
| *C. sp.* | *C. sp.* | Kakaban, Berau, Kalimantan Timur, Indonesia | MF742142 | - | * | Abboud et al. 2018 |
| *C. sp.* | *C. sp.* | Haji Buang, Maratua, Berau, Kalimantan Timur, Indonesia | MF742143 | - | * | Abboud et al. 2018 |

(*Continued*)

**Table 1.** (Continued)

| Species | Reported species upon sequence publication | Locality | COI | 16S | Dataset Used in | Source |
|---|---|---|---|---|---|---|
| | | | Accession | Accession | | |
| *C.* sp. | *C.* sp. | Haji Buang, Maratua, Berau, Kalimantan Timur, Indonesia | MF742148 | - | * | Abboud et al. 2018 |
| *C.* sp. | *C.* sp. | Danau Hidden Gam, Papua | MF742150 | - | * | Abboud et al. 2018 |
| *C.* sp. | *C.* sp. | Danau Hidden Gam, Papua | MF742151 | - | * | Abboud et al. 2018 |
| *C.* sp. | *C.* sp. | Mascot Channel, New Ireland, Papua New Guinea | MF742165 | - | * | Abboud et al. 2018 |
| *C.* sp. | *C.* sp. | Ongael Lake, Koror State, Palau | MF742183 | - | * | Abboud et al. 2018 |
| *C.* sp. | *C.* sp. | Lake Alexander, Northern Territory, Australia | MF742190 | - | * | Abboud et al. 2018 |
| *C.* sp. | *C.* sp. | Lake Alexander, Northern Territory, Australia | MF742191 | - | * | Abboud et al. 2018 |
| *C.* sp. | *C.* sp. | Lake Alexander, Northern Territory, Australia | MF742192 | - | * | Abboud et al. 2018 |
| *C.* sp. | *C.* sp. | Mascot Channel, New Ireland, Papua New Guinea | MF742209 | - | * | Abboud et al. 2018 |
| *C.* sp. | *C.* sp. | Mascot Channel, New Ireland, Papua New Guinea | MF742212 | - | * | Abboud et al. 2018 |
| *C.* sp. *1* | *C.* sp. *1* | Port Douglas, Queensland, Australia | AY319471 | - | * | Holland et al. 2004 |
| *C.* sp. *2* | *C.* sp. | Papua New Guinea | MF742198 | - | * | Abboud et al. 2018 |
| *C.* sp. *2* | *C.* sp. | Papua New Guinea | MF742199 | - | * | Abboud et al. 2018 |
| *C.* sp. *2* | *C.* sp. *2* | Observation Point, Papua New Guinea | AY319459 | - | * | Holland et al. 2004 |
| *C.* sp. *2* | *C.* sp. *2* | Observation Point, Papua New Guinea | AY319460 | - | * | Holland et al. 2004 |
| *C.* sp. *3* | *C.* sp. | Kagoshima Bay, Nagasuiro, Japan | MF742162 | - | * | Abboud et al. 2018 |
| *C.* sp. *3* | *C.* sp. | Nggatokae Mangroves, Western Solomon Islands | MF742189 | - | * | Abboud et al. 2018 |
| *C.* sp. *3* | *C.* sp. *3* | Emona, Papua New Guinea | AY319461 | - | * | Holland et al. 2004 |
| *C.* sp. *3* | *C.* sp. *3* | Emona, Papua New Guinea | AY319462 | - | * | Holland et al. 2004 |
| *C.* sp. *3* | *C.* sp. *3* | Wedding Chapel, windward O'ahu, Hawai'i, USA | AY331594 | - | * | Holland et al. 2004 |
| *C.* sp. *3* | *C.* sp. *3* | Kualoa Ranch, windward O'ahu, Hawai'i, USA | AY331595 | - | * | Holland et al. 2004 |
| *C. xamachana* | *C. xamachana* | eDNA Key Largo, FL, USA | - | MT709260 | # | Ames et al. 2021 |
| Unverified *Cassiopea* sequence | *C. andromeda* | eDNA Key Largo, FL, USA | - | MT709258 | # | Ames et al. 2021 |
| *Mastigias papua* | *Mastigias papua* | Mekeald Lake, Palau | KU901434 | KY610621 | *^ | Swift et al. 2016 |
| *Versuriga anadyomene* | *Versuriga anadyomene* | Beibu Gulf, South China Sea | KX904853 | KX904852 | *^ | Sun et al. 2019 |

Isolates with the name *C. andromeda*\* were collected as *C. xamachana* then redefined as *C. andromeda* in Holland et al. 2004 but as of this writing appear in GenBank as *C. xamachana*. Used In: * denotes *COI* alignment

^ denotes *16S* alignment

+ denotes *COI* haplotype network

# denotes *16S* haplotype network.

SH-aLRT (approximate likelihood ratio test) and 1000 non-parametric bootstraps under the Mega X model tester suggested model (TN93 + G) and edited in Figtree v1.4.

**Haplotype networks.** All 55 *COI* sequences from this study and 29 additional GenBank *COI* sequences belonging to *C. xamachana* and *C. andromeda* were used to generate a *COI*

haplotype network. 38 *16S* sequences from this study and 7 additional GenBank *16S* sequences belonging to *C. xamachana* and *C. andromeda* were used to generate a *16S* haplotype network. Two additional *16S* sequences produced from water samples in the Florida Keys for eDNA purposes (from Ames et al. 2021) were included in the *16S* haplotype map as these represented the only *Cassiopea 16S* data from the Florida Keys outside of the sequenced *C. xamachana* genome (see Table 1 for a complete list of sequences used). Haplotype networks were built using PopART v1.7 [28].

**Morphometric data.** *Cassiopea* bell diameter were recorded for each medusa and compared using a one-way analysis of variance of each of these features between sites and between mitotypes performed in R v. 4.0.3.

## Results

### Genetic identity

Of the 55 *Cassiopea COI* sequences collected in the Florida Keys, 49 had a *C. xamachana* mitochondrial haplotype and six had a *C. andromeda* haplotype, as determined by agreement with published *C. xamachana* and *C. andromeda* mitogenomes [21, 22]. The *COI* divergence between the two species was approximately ~7%, as previously reported [13, 22], with no intraspecific divergence within the *C. xamachana* or *C. andromeda* collected (Fig 1).

The *16S* and *COI* combined dataset, with sequences from GenBank, and rooted using sequences of *M. papua* and *V. anadyomene*, showed *C. xamachana* and *C. andromeda* as well supported sister taxa (posterior probability: 1, aLRT: 100%, bootstrap: 100%), and together as sister to a low supported clade that contains *C. ornata* from the Western Pacific, *C.* sp 1 from Australia, *C. culionensis*, *C.* sp 2 and *C. mayeri* from the Western Pacific. *C.* sp. 3 (from Papua New Guinea and Hawai'i) and *C. frondosa* are at the base of the tree (posterior probability: 1, aLRT: 99.5%, bootstrap: 100%) (Fig 1). Fourteen sampled individuals of undescribed *Cassiopea* from the Western Pacific (Abboud et al. 2018) remain external to known *Cassiopea* taxa. Tree topology is in agreement with previous phylogenies [3, 13].

The clade of *C. xamachana*, as defined by the published genome, is composed by sequences from the Florida Keys and Atlantic Panama, and one isolate from Palau (reported by Abboud et al. 2018). According to the analyzed dataset, except for the single Palau sequence, the *C. xamachana* mitotype is restricted to the West Atlantic (Fig 2A). *COI* sequences identified as "*C. frondosa*" and collected in Panama (Atlantic), also fell within the *C. xamachana* clade. A third common species, *C. andromeda*, includes sequences from specimens collected in Hawai'i, Mexico, Bermuda, Brazil, Florida and the Red Sea (Fig 2A & 2B).

*C. xamachana* and *C. andromeda* had ~3.1% divergence in *16S* sequences, lower than that calculated for *COI*. The six *16S* sequences from collected *C. andromeda* showed no intraspecific diversity. The 32 *C. xamachana 16S* sequences showed low intraspecific diversity (d = 0.0006) (Fig 3B).

When considering the *COI* dataset used for haplotype network building, *C. andromeda* showed 11 total haplotypes and *C. xamachana* a single haplotype (Fig 3A). *C. xamachana* showed three *16S* haplotypes, and *C. andromeda* two, though only one was from specimens collected in this study (Fig 3B). The most common *C. xamachana 16S* haplotype was present in both the Keys and Panama. The second largest *16S* haplotype was present only in upper Keys sites, in conjunction with a single individual from Big Pine Key showing a single nucleotide change. In the *16S* haplotype network, there is one intermediate between *C. xamachana* and *C. andromeda* individuals, however this sequence was collected as part of an eDNA project and maynot represent a mitochondrial haplotype present in the system [30].

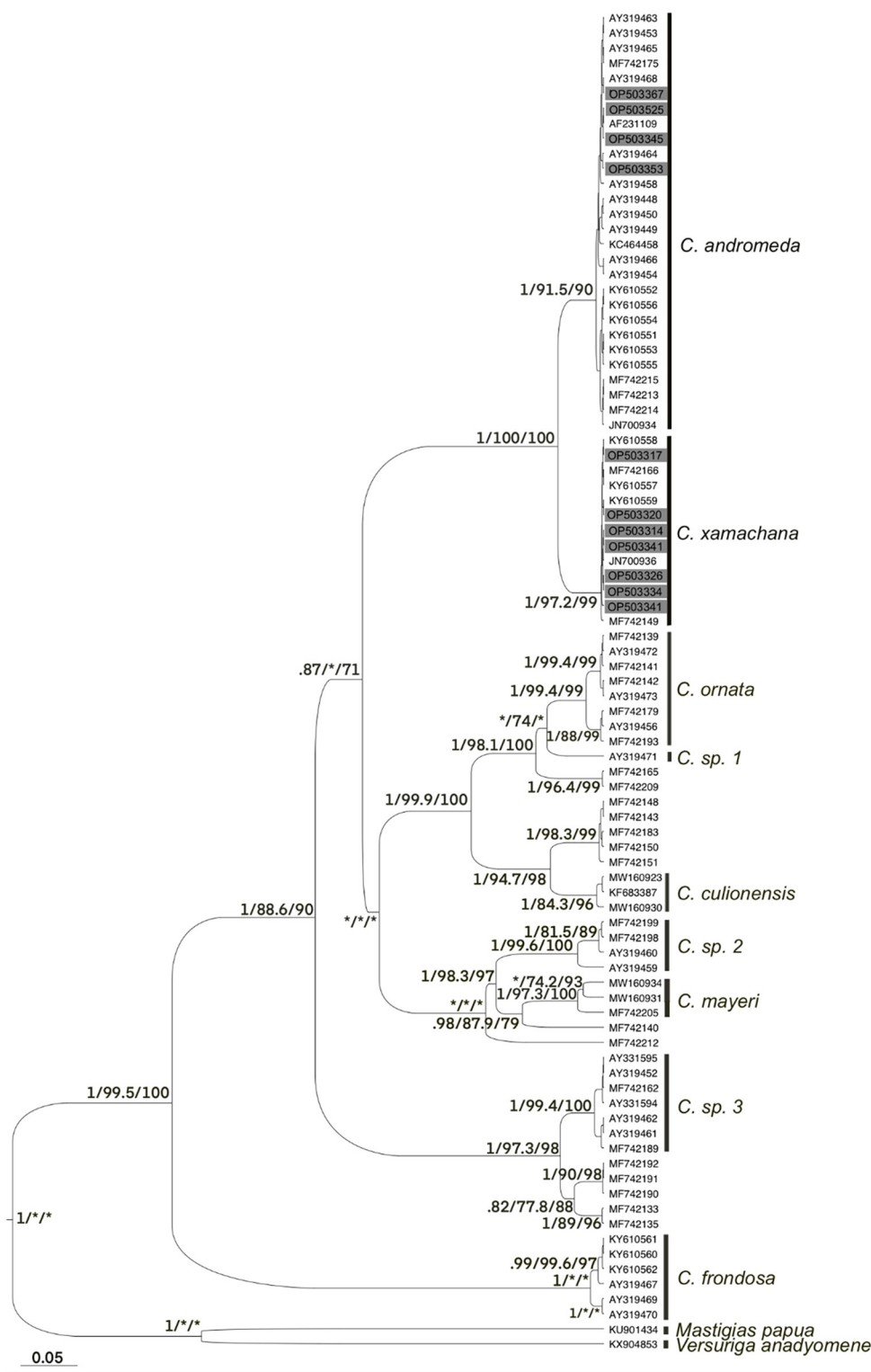

**Fig 1. Combined *COI* and *16S* tree of *Cassiopea*.** Maximum likelihood *COI* + *16S* tree with posterior probability, aLRT, and bootstrap supports. Individuals in grey were sequenced in this study. All accession numbers are for *COI*, see *16S* accession numbers in Table 1. Asterisk (*) represents under .70 (PP) or 70 (aLRT and bootstrap) support. Species names are assigned in accordance with Gamero-Mora et al. 2022.

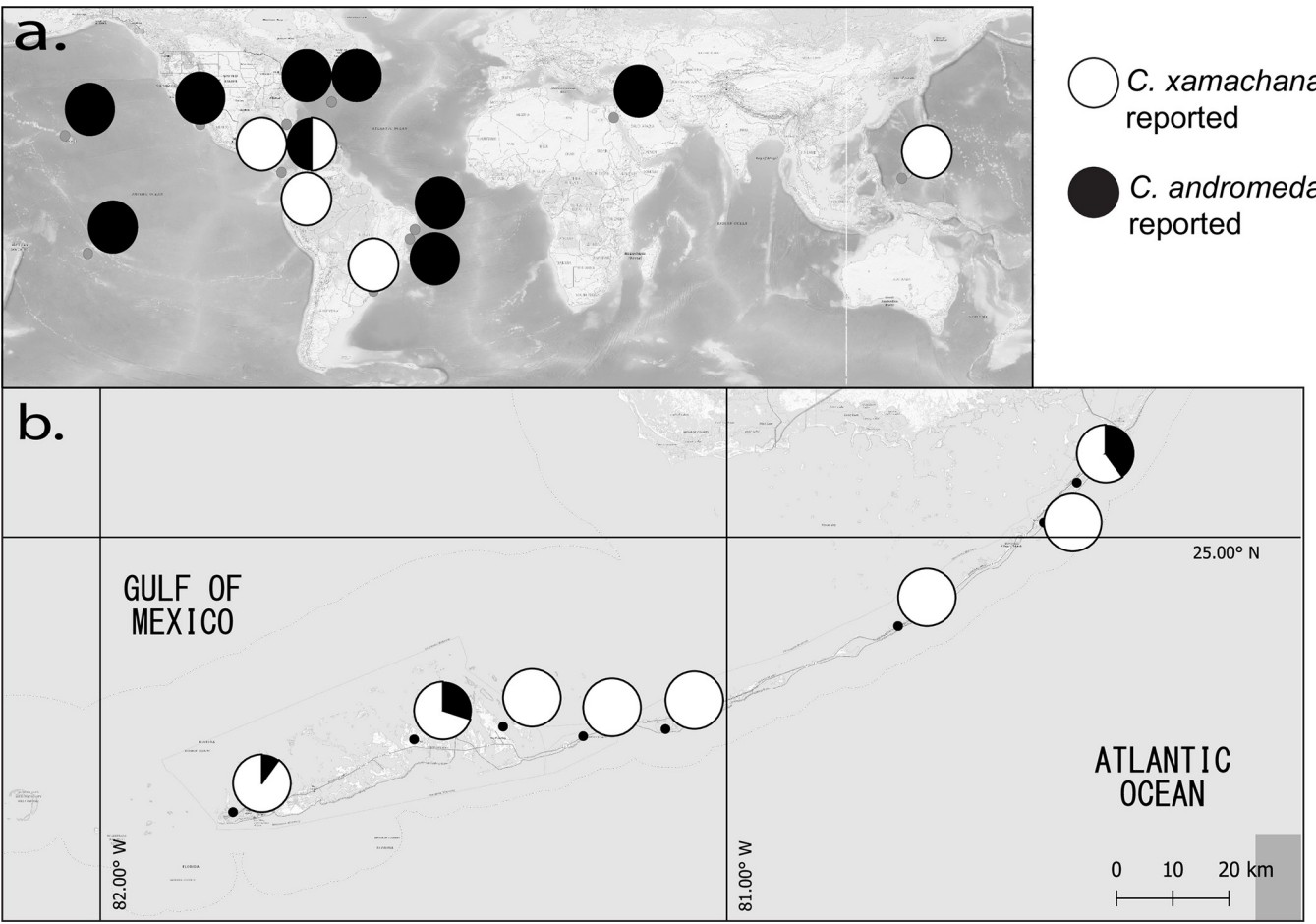

**Fig 2. Distribution of *C. xamachana* and *C. andromeda*. (a)** Global *C. andromeda* (black) and *C. xamachana* (white) distributions from sequences published in relevant scyphozoan or *Cassiopea* specific phylogenies from 2004–2022 [3, 6, 13, 17, 29]. Locations with both recorded as black-and-white. **(b)** *C. andromeda* (black) and *C. xamachana* (white) isolates from the Florida Keys from this study, pie chart indicates proportion of specimens that were *C. xamachana* and *C. andromeda* at each site.

The nuclear *28S* sequences (n = 18) from all sampled *Cassiopea* showed polymorphism, and tree topology was incongruent with mitochondrial genes sequenced (S1 Fig in S1 File). Specifically, sequences belonging to specimens collected in this study identified as *C. xamachana* and *C. andromeda* showed no differentiation. Genbank sequences of *C. andromeda* from Baja California (Genbank Acc. KY611005-7) included gaps not found in the sequences from the Keys and were closely related to two GenBank sequences of "*C. frondosa*" from Panama (likely *C. xamachana*) (Genbank Acc. KY611002-3). *C. ornata* from Palau and true *C. frondosa* from Key West (GenBank Acc. HM194838 and HM194872) were divergent from the *C. xamachana*/*C. andromeda* clade and each other.

## Geographic distribution of *C. andromeda* and *C. xamachana* within the Florida Keys

We collected 55 samples in 8 localities along the Florida Keys (S1 Table in S1 File). With 49/55 individuals, *C. xamachana* was more frequently found in samples (Fig 2B). *C. xamachana* was found in all sites and *C. andromeda* in three of eight sampling sites. In the three sites that hosted the two species, *C. xamachana* was more abundant than *C. andromeda* (proportion of

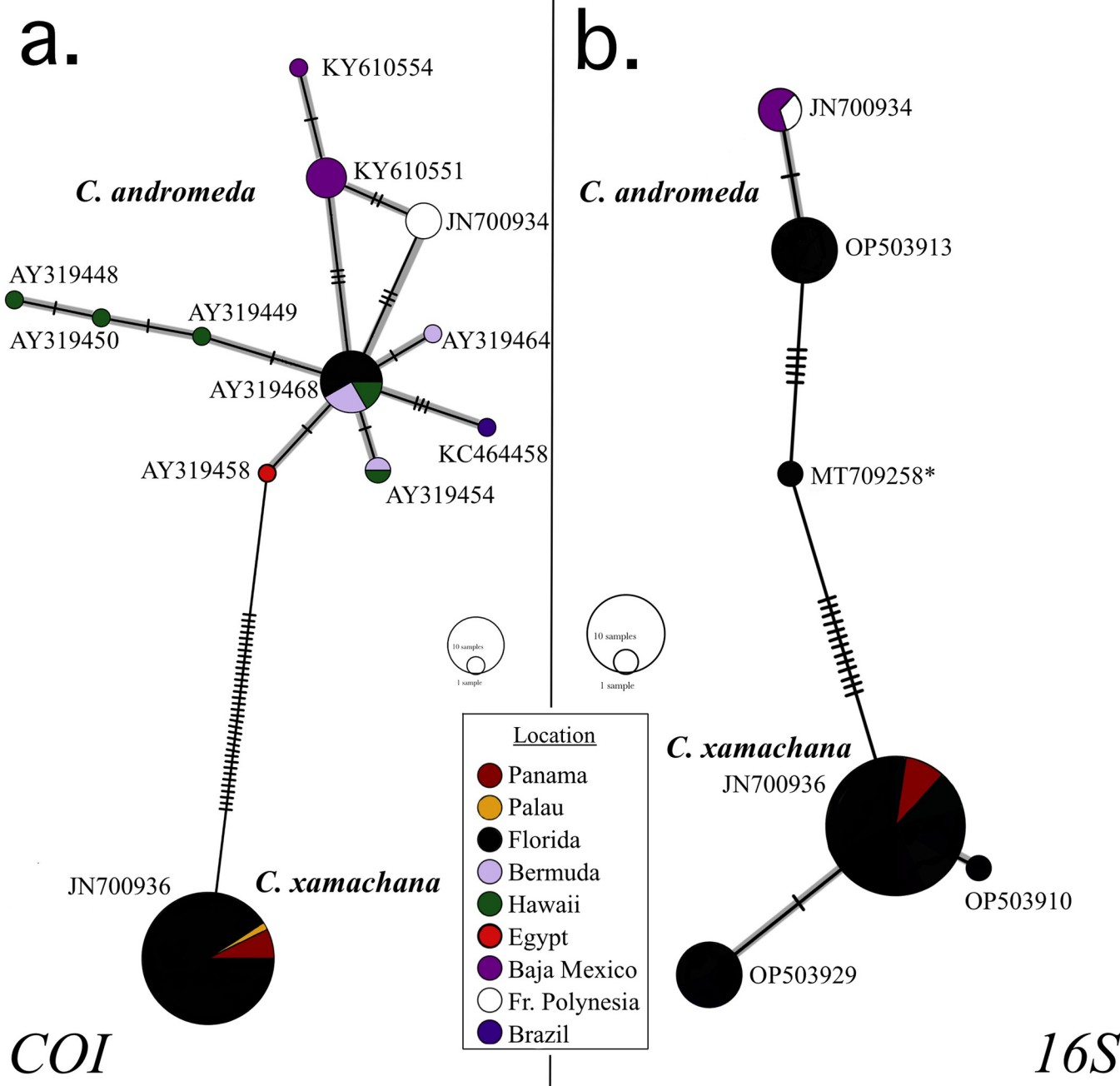

**Fig 3. Haplotype network for *C. xamachana* and *C. andromeda*. (a)** *COI* haplotype network and **(b)** *16S* haplotype network. One accession number that is consistent with the haplotype is displayed next to each group. Sequence MT709258* is a product of eDNA work and may not represent a genuine haplotype. Grey highlight connects all sequences from each species.

*C. andromeda*: 3/10 on Cudjoe Key, 1/10 on Key West, 2/5 on Key Largo). Two of the five monospecific *C. xamachana* sites were shallow lagoons (< 1m depth). The other two were low coverage tidal oceanic sites with low density and large individuals. The three sites of cohabitation were densely populated sites (>10 medusae/m$^2$) with calm water but direct oceanic exposure. Both the Key Largo site and the Key West site were at marinas.

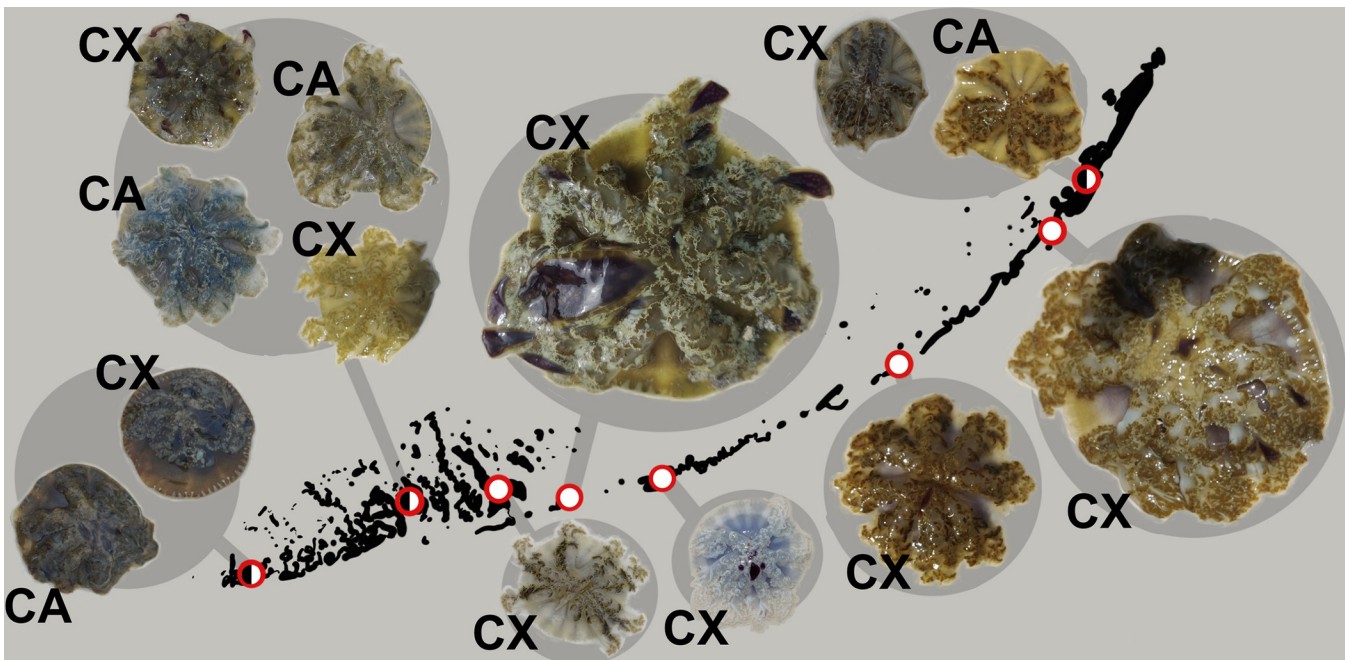

**Fig 4. Color and morphological variability in collected *Cassiopea* by site in the Florida Keys.** "CX" for *Cassiopea xamachana* and "CA" for *Cassiopea andromeda*.

*C. andromeda* have been found on both the Atlantic and Gulf sides of Key Largo, as well as on the Gulf side of Cudjoe Key and Key West.

## Size and color

At each location, *C. andromeda* and *C. xamachana* presented the same color type and similar morphology (see Supplementary Materials), with no apparent character that could distinguish between the two species. Overall, individuals with the *C. andromeda* mitotype had somewhat smaller diameters (mean = 4.8 cm) than *C. xamachana* (mean = 7.8 cm) but not to a significant degree (ANOVA, $F (2,47)$ = 3.69, p = 0.061). Two of the sites where *C. andromeda* were found (Cudjoe Key and Key Largo Ocean Bay Marina) had primarily small individuals, and site was the most important factor in size determination (ANOVA, $F (1,47)$ = 5.08, p = 0.029). As all specimens were preserved in ethanol as opposed to formalin, no in-depth comparative morphological analyses were performed. A general inspection showed that *C. andromeda* (n = 2) and *C. xamachana* (n = 3), when devoid of symbionts, had different bell markings, however, with such a limited dataset, no conclusions could be drawn (S2 Fig in S1 File). All specimen photographs are included in supplementary materials.

With regard to coloration, *Cassiopea* collected from this work were blue, white, purple, pink, brown, and green. This is a well-known phenomenon in *Cassiopea*, and the exact dynamics of color are poorly understood. Color profiles were generally consistent within sites but were highly variable between sites (Fig 4). There was no consistent difference in color bell, oral arm or paddle coloration between species within each site at which the mitotypes were cooccurring.

## Discussion

*C. xamachana*, has been genetically undersampled in the Florida Keys relative to its use in research and laboratory work [11, 14, 21, 31]. A belief that *C. xamachana* was a *C. andromeda*

subpopulation likely contributed to this lack of focus on dense geographic sampling [3, 6]. Moreover, in the last 20 years *C. xamachana COI* isolates have occasionally been misidentified as *C. frondosa*, a very distantly related *Cassiopea*, further adding to the taxonomic confusion. Our data support the notion that *C. xamachana* and *C. andromeda* mitotypes are now both found in the Florida Keys. Although mitochondrial DNA suggests these two species are reciprocally monophyletic clades, additional nuclear genes are necessary to confirm their monophyly. The *Cassiopea* collected here were not distinguishable in the field and they inhabited the same shallow water, sometimes coexisting side by side in the same location. 89% of our samples matched *C. xamachana*. *C. xamachana* was also sampled in 100% of sampling sites (8/8), while *C. andromeda* was sampled in 38% (3/8). We thus show that in our sampling effort, the *C. xamachana* mitotype is more abundant both in terms of number of jellyfish and locations where it is found.

Our results also confirm some findings of recent eDNA analyses conducted in the same area that recorded *16S* residues of both species in the Florida Keys [30]. While we find the *C. xamachana* and *C. andromeda* mitotypes in our data, we fail to find the intermediate *16S* signature found by Ames et al. 2021, a sequence that may have been an interspecific chimera. Despite having more representatives in this study, there was no diversity within the *C. xamachana COI* sequences and little diversity in their *16S* profiles. This may represent a higher degree of continuity across the Florida Keys and Panama than expected. Further study of the exact boundaries that impact *Cassiopea* genetic populations is needed. While *C. xamachana* has been found in Brazil and Palau [6, 32], it does not yet have the non-native range demonstrated by the *C. andromeda* clade.

While there has been no demonstrated divergence in behavior or tolerance to environmental factor between *C. xamachana* and *C. andromeda*, there has also been no study formally comparing them. Our data show that wild-caught *Cassiopea*, even from single locations present clear hazards for comparative analysis if not properly identified [4]. This brings into foreground the inadvisability of treating results from Floridian samples and other locations as representatives of one clade without genetic evidence. In addition to genetic study, the Keys population would benefit from careful morphological and ecological analysis within mixed assemblage sites to parse whether these cooccurring populations have distinct diagnostic morphometrics or ecological features.

The mitochondrial markers analyzed in this work present evidence of historical genetic separation between *C. xamachana* and *C. andromeda*, the nuclear marker (*28S*), however, does not. As *Cassiopea* has very few published *28S* sequences, some of which certainly suffer from the same issues of misidentification as the *COI* isolates, firm conclusions cannot be drawn as to the usefulness of *28S* for species delimitation. Given the low mitochondrial divergence relative to a 10% benchmark [1], the *28S* results may indicate introgression and hybridization between *C. andromeda* and *C. xamachana*. In-depth study of a larger array of nuclear markers is needed to parse the hybridization potential or degree of *C. xamachana* and *C. andromeda* within the Florida Keys.

As only 11% of individuals found in the Keys in this sampling effort were *C. andromeda*, only relatively dense sampling (either large collections from multiple locations or environmental sequencing) could identify the mix. In Brazil, multiple *Cassiopea* invasions have resulted in both *C. xamachana* and *C. andromeda* populations, but these were found in separate sampling efforts [17, 29, 32]. Locations with *Cassiopea*, especially those with a paucity of sequences, may present the sort of assemblages already identified here in the Florida Keys, Hawai'i, Brazil, the Philippines and Palau, and may require multiple rounds of sampling to parse [3, 6, 13]. Some Pacific *Cassiopea* populations remain unidentified (see the sequences without species identity in Fig 1), further hampering our understanding of invasion history within the genus. Greater

sampling numbers are needed to further characterize species distributions within the Keys and elsewhere.

Finally, in constructing the phylogeny, we found instances of *Cassiopea* misidentification in GenBank. Three *COI* sequences identified as *Cassiopea frondosa* were instead *C. xamachana*. Additionally, five sequences identified as *C. xamachana* were instead *C. andromeda*, despite their originating text correctly identifying their species affinity [3]. This accounts for five of sixteen total identified *C. andromeda* sequences. This indicates a general problem with Gen-Bank sequences and is a result of the taxonomic confusion that has surrounded *Cassiopea* species.

## Conclusion

*Cassiopea* in the Florida Keys has long been defined as two species, the deeper-water, distinctive *C. frondosa* and the shallow-water *C. xamachana* [12]. In 1960 and again in 2004, *C. xamachana* was relegated to a junior synonym of *C. andromeda* [3, 19]. Using a phylogenetic approach, we show that *C. andromeda* and *C. xamachana* mitochondrial genotypes are both found in sympatry in the Florida Keys, showing no obvious morphological differences. We show that it is difficult to determine the population history of *Cassiopea* collected in shallow water in the Keys without proper molecular barcoding. This is relevant because a wealth of research has been performed with various *Cassiopea* without a proper assessment of the species it was conducted on. We also found evidence that *Cassiopea* research has suffered from frequent species misidentification. This paper calls for deeper sampling of jellyfish assemblages within *Cassiopea* and other highly cryptic scyphozoan genera. It also indicated that a proper species identification that involves molecular barcoding is essential for any work on *Cassiopea*, especially from Florida. This even more crucial as *Cassiopea* continues to successfully establish itself as an emerging model system for physiological studies and as a proxy for investigations on zooxanthellae-Cnidaria interaction. Caution should be exercised in generalizing result from published studies that assumed *Cassiopea* identity without explicitly investigating species identification with molecular tools.

## Supporting information

**S1 File. Includes S1 and S2 Figs, S1 and S2 Tables.**
(PDF)

## Acknowledgments

We thank the Florida Fish and Wildlife Conservation Commission for their help and guidance.

## Author Contributions

**Conceptualization:** Kaden Muffett.

**Data curation:** Kaden Muffett.

**Formal analysis:** Kaden Muffett.

**Funding acquisition:** Maria Pia Miglietta.

**Investigation:** Kaden Muffett.

**Methodology:** Kaden Muffett.

**Supervision:** Maria Pia Miglietta.

**Validation:** Maria Pia Miglietta.

**Writing – original draft:** Kaden Muffett.

**Writing – review & editing:** Maria Pia Miglietta.

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
