## [Decision Letter · Decision Letter 0]

6 Jan 2023

PONE-D-22-31726Demystifying Cassiopea species identity in the Florida Keys: Cassiopea xamachana and Cassiopea andromeda coexist in shallow watersPLOS ONE

Dear Dr. Kaden Muffett

Thank you for submitting your manuscript to PLOS ONE. After careful consideration, we feel that it has merit but yet does not fully meet PLOS ONE’s publication criteria as it currently stands. Therefore, we invite you to submit a revised version of the manuscript that addresses the points raised during the review process. I received feedback from three reviewers and both were very positive about your manuscript. In this way, I indicate that your manuscript can be accepted soon after making the specific modifications indicated by the reviewers. Please review the points listed and send us an updated version of your manuscript.

We look forward to receiving your revised manuscript.

Kind regards,

Sergio N. Stampar, Dr.

Academic Editor

PLOS ONE

Journal Requirements:

"We thank the NSF and NOAA Texas Sea Grant for funding this work."

"This work was supported by the National Science Foundation (NSF IOS #1936565) to author MPM. This work was also supported by the Texas Sea Grant to author KMM.

6. We note that Figures 3 and S1 in your submission contain [map/satellite] images which may be copyrighted. All PLOS content is published under the Creative Commons Attribution License (CC BY 4.0), which means that the manuscript, images, and Supporting Information files will be freely available online, and any third party is permitted to access, download, copy, distribute, and use these materials in any way, even commercially, with proper attribution. For these reasons, we cannot publish previously copyrighted maps or satellite images created using proprietary data, such as Google software (Google Maps, Street View, and Earth). For more information, see our copyright guidelines: http://journals.plos.org/plosone/s/licenses-and-copyright.

a) You may seek permission from the original copyright holder of Figures 3 and S1 to publish the content specifically under the CC BY 4.0 license.  

Natural Earth (public domain): http://www.naturalearthdata.com/.

7. Please include a separate caption for each figure in your main manuscript.

Additional Editor Comments:

Dear Dr. Kaden Muffett,

I received feedback from three reviewers and both were very positive about your manuscript. In this way, I indicate that your manuscript can be accepted soon after making the specific modifications indicated by the reviewers. Please review the points listed and send us an updated version of your manuscript.

Reviewers' comments:

Reviewer's Responses to Questions

**Comments to the Author**

1. Is the manuscript technically sound, and do the data support the conclusions?

Reviewer #1: Yes

Reviewer #2: Yes

Reviewer #3: Yes

2. Has the statistical analysis been performed appropriately and rigorously? 

Reviewer #1: Yes

Reviewer #2: Yes

Reviewer #3: Yes

3. Have the authors made all data underlying the findings in their manuscript fully available?

Reviewer #1: Yes

Reviewer #2: Yes

Reviewer #3: Yes

4. Is the manuscript presented in an intelligible fashion and written in standard English?

Reviewer #1: Yes

Reviewer #2: Yes

Reviewer #3: Yes

5. Review Comments to the Author

Reviewer #1: dear authors,

the topic of your paper is very interesting and the paper opens a lot of questions on the species identification in that location where two different species coexist as well as past taxonomic misidentifications. All presented results seems to support the conclusion even if sometimes are a litlle bit confusing as stated along the manuscript file pdf attached.

all data of the manuscript are available. Please see the file attached for detailed comments

regards

Reviewer #2: General comments:

This is a very interesting paper about identifying species of Cassiopea in Florida. The study highlights the cryptic nature of Cassiopea and the importance of using molecular work for species identification. Additionally, it confirms that multiple species of Cassiopea can co-exist in the same area, and so location should not be used as a factor to identify species.

Overall, the paper is well written, and represents the results in a way that is easy to understand and interpret. However, I do have a couple of comments and questions that should be addressed in the paper. Mainly, the authors conclude that C. andromeda and C. xamachana are different species based off genetics and morphology. However, the authors also state that they did not complete any detailed morphological comparison between the two species. To be able to definitively conclude that there are two species occurring, especially as they so closely related, a detailed morphological comparison needs to be done before making such a conclusion. As a result, the authors need to tone back their conclusion that they are definitely different species and include another paragraph in the discussion discussing this further, along with strengthening their argument about why the authors concluded that there are two species based on genetics.

Specific comments:

Abstract:

Line 10: “upset multiple times” is unclear terminology. “Confused” may be a better term to use here

Introduction:

Line 26: “sampled broadly but shallowly” is confusing terminology. Do the authors mean shallow depth here? What depths do they define as shallow? These references refer to Cassiopea, which are typically found in shallow environments. Aurelia have been collected at depths of up to 18m though and so a clearer way of describing this could be close to shore.

Line 30: What other scyphozoan lineages?

Line 44: Is C. xamachana morphologically distinct from C. andromeda? They have often been confused taxonomically because they have similar morphological characteristics. Additionally, this is conflicting with the rest of the paper.

Line 61: I think the authors need to clarify that they did not look at deeper depths, where C. frondosa are known to occur. Cite the Fitt et al. 2021 paper.

Methods:

Line 69: What was measured? Their diameter?

Line 70: why were the samples put in ethanol rather than formalin for morphological analysis?

Results:

Line 136: Need references for previously published mitogenomes

Line 152: There are other sequences of C. xamachana on Genbank that are not in the Caribbean. Please see Stampar et al (2021) The puzzling occurrence of the upside-down jellyfish Cassiopea (Cnidaria: Scyphozoa) along the Brazilian coast: a result of several invasion events? And Gamero-Mora (2019) Regenerative capacity of the upside-down jellyfish Cassiopea xamachana.

Line 188: How shallow?

Line 190: How did the authors measure the density for each site? This is not included in the methods.

Line 194: Deeper sampling? What depths do they mean by this? Cassiopea frondosa has been sampled at deeper depths within the area so this needs to be clarified

Line 201: These stats for the difference between bell diameters needs to be included in the methods

Line 204: If no comparative morphological analysis could be done, line 198 needs to be revised. The authors shouldn’t state that there were no morphological differences between the two species if no detailed comparison was done

Line 211: A bit more detail needs to be provided here. Did the colour vary between species as well as between sites?

Discussion:

Line 214: Remove “dramatically”

Line 216: Remove “only”

Line 221: Hard to conclude that C. andromeda and C. xamachana are different species when they are in the same monophyletic group and this study did not do any morphological comparisons, but state that there was no morphological difference between species. This sentence needs to be revised as such as I don’t believe the authors can confidently state that C. andromeda and C. xamachana are different species without a proper morphological comparison. If the authors are going to state there are no morphological differences between the species, references need to be included to support this.

Expanding on above, I think the authors need to include a paragraph discussing how a proper morphological comparison needs to be done to support their findings that genetics shows that two species are occurring in Florida.

Line 265: Again, the authors state that there were no morphological differences between the two species, but have also acknowledged that they didn’t do a detailed morphological comparison.

Reviewer #3: The manuscript is well written, clear and easy to understand. Below are some specific observations/ suggestions.

Line 15 – Scientific names are commonly written out in full when they first appear in the abstract.

Lines 31, 42, 94, etc. – The names are again written out in full when they first appear after the abstract and are then abbreviated upon further use.

Line 43 -- Ecotype could replaced by species or taxon.

Line 65 – “through agreement >98% to either genome.”, which is the reference to those genomes?

Line 112 – For the multi-gene analysis (16S+COI), do the authors allowed individual models for the different genomic loci (genes or codon positions)? Why does the 28S dataset was not included in the combined analysis for tree generation?

Lines 140, 154, 160, 228, etc. – Mastigias papua, Versuriga anadyomene, Cassiopea andromeda, Cassiopea xamachana can be abbreviated (they were written out in full previously).

Some scientific names appear abbreviated at the beginning of sentences (e.g., line 176) while others were written out in full at the beginning of sentences (e.g., line 162), I suggest homogenizing the format.

Figure 3b could be improved by adding information about latitude and longitude (e.g., by adding, to the figure, an upper layer with a coordinate grid) and using a bigger font size for “Gulf of Mexico” and “Florida Keys ”

Figure 4 could be improved by coloring with two colors the circles that indicate the locations where both species were recorded (black and white as in figure three).

6. PLOS authors have the option to publish the peer review history of their article (what does this mean?). If published, this will include your full peer review and any attached files.

Reviewer #1: No

Reviewer #2: No

Reviewer #3: No

---

## [Author Response · Author response to Decision Letter 0]

20 Jan 2023

Dear editor and reviewers,

 Thank you for reviewing this manuscript, your thoughtful comments have improved the text substantially. In this revised draft we have made the following high-level changes to the manuscript overall:

1. The manuscript formatting, figures, etc. have been revised to comply with journal guidelines.

2. The abstract, results and discussion section have been reworked to emphasize the lack of clarity that still exists on the C. andromeda/C. xamachana species boundary, and the necessity for more ecological and morphological analysis of the medusae. 

Specific line edit responses are listed below. We hope that you find the revisions in line with expectations. 

Thank you again,

KM Muffett

Reviewer #1: 

Dear authors,

The topic of your paper is very interesting and the paper opens a lot of questions on the species identification in that location where two different species coexist as well as past taxonomic misidentifications. All presented results seem to support the conclusion, even if sometimes they are a little bit confusing as stated along the manuscript file pdf attached.

 Thank you for your kind words and your support of this manuscript.

1. Line 14: Is Cassiopea present beyond the Florida Keys?

Cassiopea extends beyond the Florida Keys into southern Florida with limited accounts on the panhandle (northern Fl).

2. Line 68: Are S1 and S2 supplementary figures?

S1 and S2 are supplementary figures, a description of all supplementary figures have been added at the end of the text.

3. Line 78: Why were there fewer 16S and 28S genes sequenced?

We identified clades with the 16 dataset, and selected representatives of each clade for further 28S sequencing. This allowed for the most cost/time effective approach. However, we also encountered difficulties in amplifying the 28S sequences. This resulted in having fewer 28S sequences than 16S.

4. Line 85: How did you assign sequence identity?

Sequence identity was assigned through agreement with either the published Florida Keys C. xamachana genome or French Polynesian C. andromeda mitogenome. This has now been specified in the text.

5. Move long table

Unfortunately, table one positioning must follow the first paragraph it is mentioned to be in line with journal guidelines.

6. How many sequences are included in the combined 16S and COI dataset? Did you use this to build fig 1?

The combined COI and 16S dataset includes 89 COI and 22 16S sequences. All sequences are described in the “Combined COI and 16S Cassiopea Phylogeny Dataset” section of the methodology. Figure 1’s caption has been updated to make its provenance clearer.

7. Results section clarity

We have endeavored to improve the clarity of the results section. 

8. Line 136: Did you calculate the divergence between species?

Yes, specified to clarify that this is in reference to COI divergence.

9. Line 152: Cite figure 2 before fig 3, check figure references in text

All figures have been reorganized and figure references have been checked and corrected.

10. Move speculative sentences to the discussion.

This has been corrected as specified.

11. Line 227: Rephrase statement on Cassiopea COI identity.

The statement as presented is correct, within Cassiopea xamachana COI sequences there was no nucleotide diversity.

12. Line 244: This sentence is unclear, rephrase

This has been rephased for clarity.

13. Add comments on unassigned Cassiopea sequences from fig 1 in discussion section

This has been corrected as specified.

14. Check Gamero-Mora 2022 citation, Ohdera 2019 and Deidun 2018 citation

This has been corrected as specified.

 

Reviewer #2

This is a very interesting paper about identifying species of Cassiopea in Florida. The study highlights the cryptic nature of Cassiopea and the importance of using molecular work for species identification. Additionally, it confirms that multiple species of Cassiopea can co-exist in the same area, and so location should not be used as a factor to identify species.

Overall, the paper is well written, and represents the results in a way that is easy to understand and interpret. However, I do have a couple of comments and questions that should be addressed in the paper. Mainly, the authors conclude that C. andromeda and C. xamachana are different species based off genetics and morphology. However, the authors also state that they did not complete any detailed morphological comparison between the two species. To be able to definitively conclude that there are two species occurring, especially as they so closely related, a detailed morphological comparison needs to be done before making such a conclusion. As a result, the authors need to tone back their conclusion that they are definitely different species and include another paragraph in the discussion discussing this further, along with strengthening their argument about why the authors concluded that there are two species based on genetics.

Thank you for your support and constructive criticism of the manuscript. We have revised claims on species identity to bring them more into line with the uncertain nature of the data, and added a call for detailed morphological and ecological study in the discussion section.

1. Line 10: “upset multiple times” is unclear terminology. “Confused” may be a better term to use here

“Upset” has been replaced with “revised”.

2. Line 26: “sampled broadly but shallowly” is confusing terminology. Do the authors mean shallow depth here? What depths do they define as shallow? These references refer to Cassiopea, which are typically found in shallow environments. Aurelia have been collected at depths of up to 18m though and so a clearer way of describing this could be close to shore.

Thank you, “broadly but shallowly” here meant sampled across a large geographic region but with limited specimen numbers at each location. This has been replaced so as to make the text clearer.

3. Line 30: What other scyphozoan lineages?

Possible additionally lineages may be near-shore invaders like Phyllorhiza spp., outside of scyphozoa, genera like Turritopsis also have a history defined by cryptic and near-cryptic invasion. Without detailed study of the variation within coastal lineages like Phyllorhiza, with so many undersampled locations, diversity is impossible to ascertain. 

4. Line 44: Is C. xamachana morphologically distinct from C. andromeda? They have often been confused taxonomically because they have similar morphological characteristics. Additionally, this is conflicting with the rest of the paper.

Thank you for suggesting I clarify this. While the descriptions are certainly distinct, the traits discussed are highly variable. I have now added this line and switched the less familiar “ecotype” with the more familiar “morphotype”.

5. Line 61: I think the authors need to clarify that they did not look at deeper depths, where C. frondosa are known to occur. Cite the Fitt et al. 2021 paper.

This has now been clarified.

6. Line 69: What was measured? Their diameter?

The diameter was measured, this has now been clarified in text.

7. Line 70: why were the samples put in ethanol rather than formalin for morphological analysis?

Preserved samples were not originally intended for morphological analysis, this was an addition made later.

8. Line 136: Need references for previously published mitogenomes

This has been corrected as specified.

9. Line 152: There are other sequences of C. xamachana on Genbank that are not in the Caribbean. Please see Stampar et al (2021) The puzzling occurrence of the upside-down jellyfish Cassiopea (Cnidaria: Scyphozoa) along the Brazilian coast: a result of several invasion events? And Gamero-Mora (2019) Regenerative capacity of the upside-down jellyfish Cassiopea xamachana.

Thank you. I had originally excluded laboratory specimens, however, seeing the provenance of the colony I will now update the paper to reflect that C. xamachana is found in Brazil as well.

10. Line 188: How shallow?

Corrected. Specified as under 1m depth.

11. Line 190: How did the authors measure the density for each site? This is not included in the methods.

A m2 was marked and individuals within that area were counted. This has now been added to the collection section of the methods.

12. Line 194: Deeper sampling? What depths do they mean by this? Cassiopea frondosa has been sampled at deeper depths within the area so this needs to be clarified

Deeper here meant “greater N”. This has been replaced.

13. Line 201: These stats for the difference between bell diameters needs to be included in the methods

This has been corrected as suggested.

14. Line 204: If no comparative morphological analysis could be done, line 198 needs to be revised. The authors shouldn’t state that there were no morphological differences between the two species if no detailed comparison was done

This has been corrected as suggested.

15. Line 211: A bit more detail needs to be provided here. Did the colour vary between species as well as between sites?:

Coloration of C. andromeda mitotype individuals was similar to C. xamachana individuals at sites where they cooccurred. A note not this effect has been added.

16. Line 214: Remove “dramatically”

This has been corrected as suggested.

17. Line 216: Remove “only”

This has been corrected as suggested.

18. Line 221: Hard to conclude that C. andromeda and C. xamachana are different species when they are in the same monophyletic group and this study did not do any morphological comparisons, but state that there was no morphological difference between species. This sentence needs to be revised as such as I don’t believe the authors can confidently state that C. andromeda and C. xamachana are different species without a proper morphological comparison. If the authors are going to state there are no morphological differences between the species, references need to be included to support this.

Expanding on above, I think the authors need to include a paragraph discussing how a proper morphological comparison needs to be done to support their findings that genetics shows that two species are occurring in Florida. AND Line 265: Again, the authors state that there were no morphological differences between the two species, but have also acknowledged that they didn’t do a detailed morphological comparison.

We have added text to the discussion calling for morphological and ecological study of mixed populations to determine whether there are indeed differences, we have also walked back claims about morphology and increased emphasis on the lack of clarity in the nature of the species boundary (including presence/absence).

Reviewer #3: The manuscript is well written, clear and easy to understand. Below are some specific observations/ suggestions.

1. Line 15 – Scientific names are commonly written out in full when they first appear in the abstract.

This has been corrected as suggested.

2. Lines 31, 42, 94, etc. – The names are again written out in full when they first appear after the abstract and are then abbreviated upon further use.

This has been corrected as suggested.

3. Line 43 -- Ecotype could replaced by species or taxon.

Replaced by morphotype, as it is not clear that the foundational traits on which they were differentiated are distinctive between groups and often vary within C. xamachana and C. andromeda (e.g. Color)

4. Line 65 – “through agreement >98% to either genome.”, which is the reference to those genomes?

This has been corrected as suggested.

5. Line 112 – For the multi-gene analysis (16S+COI), do the authors allowed individual models for the different genomic loci (genes or codon positions)? Why does the 28S dataset was not included in the combined analysis for tree generation?

Individual models were used as laid out in the paragraph starting with “Models for all COI codons” (COI codon position 1: TN93+I, COI codon position 2: TN93+I, COI codon position 3: TN93+G+I, 16S: GTR+G+I). With such a limited comparative dataset (too few Cassiopea with previously published 28S sequences) addending it to the COI and 16S seemed inadvisable. 

6. Lines 140, 154, 160, 228, etc. – Mastigias papua, Versuriga anadyomene, Cassiopea andromeda, Cassiopea xamachana can be abbreviated (they were written out in full previously).

This has been corrected as suggested.

7. Some scientific names appear abbreviated at the beginning of sentences (e.g., line 176) while others were written out in full at the beginning of sentences (e.g., line 162), I suggest homogenizing the format.

This has been corrected as suggested.

8. Figure 3b could be improved by adding information about latitude and longitude (e.g., by adding, to the figure, an upper layer with a coordinate grid) and using a bigger font size for “Gulf of Mexico” and “Florida Keys ”

This has been corrected as suggested.

9. Figure 4 could be improved by coloring with two colors the circles that indicate the locations where both species were recorded (black and white as in figure three).

This has been corrected as suggested.

---

## [Decision Letter · Decision Letter 1]

9 Mar 2023

Demystifying Cassiopea species identity in the Florida Keys: Cassiopea xamachana and Cassiopea andromeda coexist in shallow waters

PONE-D-22-31726R1

Dear Dr. Muffett,

We’re pleased to inform you that your manuscript has been judged scientifically suitable for publication and will be formally accepted for publication once it meets all outstanding technical requirements.

Kind regards,

Sergio N. Stampar, Dr.

Academic Editor

PLOS ONE

Additional Editor Comments (optional):

Dear authors,

After further evaluation by the reviewers, I am pleased to inform that the manuscript is ready for publication in Plos One.

Kind regards

Sergio Stampar

Reviewers' comments:

Reviewer's Responses to Questions

**Comments to the Author**

1. If the authors have adequately addressed your comments raised in a previous round of review and you feel that this manuscript is now acceptable for publication, you may indicate that here to bypass the “Comments to the Author” section, enter your conflict of interest statement in the “Confidential to Editor” section, and submit your "Accept" recommendation.

Reviewer #1: All comments have been addressed

Reviewer #2: All comments have been addressed

Reviewer #3: All comments have been addressed

2. Is the manuscript technically sound, and do the data support the conclusions?

Reviewer #1: Yes

Reviewer #2: (No Response)

Reviewer #3: Yes

3. Has the statistical analysis been performed appropriately and rigorously? 

Reviewer #1: N/A

Reviewer #2: (No Response)

Reviewer #3: Yes

4. Have the authors made all data underlying the findings in their manuscript fully available?

Reviewer #1: Yes

Reviewer #2: (No Response)

Reviewer #3: Yes

5. Is the manuscript presented in an intelligible fashion and written in standard English?

Reviewer #1: Yes

Reviewer #2: (No Response)

Reviewer #3: Yes

6. Review Comments to the Author

Reviewer #1: dear authors, you have made the manuscript better by following the reviewers' comments. good for pubblication

Reviewer #2: (No Response)

Reviewer #3: (No Response)

7. PLOS authors have the option to publish the peer review history of their article (what does this mean?). If published, this will include your full peer review and any attached files.

Reviewer #1: No

Reviewer #2: No

Reviewer #3: **Yes: **Edgar Gamero-Mora

---

## [Editor Report · Acceptance letter]

17 Mar 2023

PONE-D-22-31726R1 

Demystifying *Cassiopea* species identity in the Florida Keys:
*Cassiopea xamachana* and *Cassiopea andromeda* coexist in shallow waters 

Dear Dr. Muffett:

I'm pleased to inform you that your manuscript has been deemed suitable for publication in PLOS ONE. Congratulations! Your manuscript is now with our production department. 

Kind regards, 

on behalf of

Dr. Sergio N. Stampar 

Academic Editor

PLOS ONE